# Dissecting the immunosuppressive tumor microenvironments in Glioblastoma-on-a-Chip for optimized PD-1 immunotherapy

Xin Cui[1,2,3†], Chao Ma[1,2†], Varshini Vasudevaraja[4], Jonathan Serrano[4], Jie Tong[1,2], Yansong Peng[2], Michael Delorenzo[4], Guomiao Shen[4], Joshua Frenster[5], Renee-Tyler Tan Morales[2], Weiyi Qian[1], Aristotelis Tsirigos[4], Andrew S Chi[6,7], Rajan Jain[8,9], Sylvia C Kurz[6,9], Erik P Sulman[6,10], Dimitris G Placantonakis[6,9], Matija Snuderl[4,6], Weiqiang Chen[1,2,6*]

[1]Department of Mechanical and Aerospace Engineering, New York University, Brooklyn, United States; [2]Department of Biomedical Engineering, New York University, Brooklyn, United States; [3]Department of Biomedical Engineering, Jinan University, Guangzhou, China; [4]Department of Pathology, NYU Langone Health, New York, United States; [5]Stem Cell Biology Program, NYU School of Medicine, New York, United States; [6]Perlmutter Cancer Center, NYU Langone Health, New York, United States; [7]Department of Neurology, NYU Langone Health, New York, United States; [8]Department of Neuroradiology, NYU Langone Health, New York, United States; [9]Department of Neurosurgery, NYU Langone Health, New York, United States; [10]Department of Radiation Oncology, NYU Langone Health, New York, United States

*For correspondence:
wchen@nyu.edu

†These authors contributed equally to this work

**Abstract** Programmed cell death protein-1 (PD-1) checkpoint immunotherapy efficacy remains unpredictable in glioblastoma (GBM) patients due to the genetic heterogeneity and immunosuppressive tumor microenvironments. Here, we report a microfluidics-based, patient-specific 'GBM-on-a-Chip' microphysiological system to dissect the heterogeneity of immunosuppressive tumor microenvironments and optimize anti-PD-1 immunotherapy for different GBM subtypes. Our clinical and experimental analyses demonstrated that molecularly distinct GBM subtypes have distinct epigenetic and immune signatures that may lead to different immunosuppressive mechanisms. The real-time analysis in GBM-on-a-Chip showed that mesenchymal GBM niche attracted low number of allogeneic CD154+CD8+ T-cells but abundant CD163+ tumor-associated macrophages (TAMs), and expressed elevated PD-1/PD-L1 immune checkpoints and TGF-β1, IL-10, and CSF-1 cytokines compared to proneural GBM. To enhance PD-1 inhibitor nivolumab efficacy, we co-administered a CSF-1R inhibitor BLZ945 to ablate CD163+ M2-TAMs and strengthened CD154+CD8+ T-cell functionality and GBM apoptosis on-chip. Our ex vivo patient-specific GBM-on-a-Chip provides an avenue for a personalized screening of immunotherapies for GBM patients.

## Introduction

Glioblastoma (GBM) is the most common and aggressive primary brain tumor among adults, with an average survival of less than 14 months despite aggressive surgery, chemotherapy, and radiotherapy (*Stupp et al., 2005*). Programmed cell death protein-1 (PD-1) checkpoint blockade has emerged as a remarkable immunotherapy in pilot GBM clinical trials, yet the durability of patient remission remains largely unpredictable due to heterogeneous tumor immune microenvironments of GBM

patients (*Cloughesy et al., 2019*; *Di Tomaso et al., 2010*). At most, 8% of GBM patients demonstrate long-term responses in ongoing trials (*Reardon et al., 2017*). However, a lack of clinical biomarkers to predict response represents a critical unmet need to identify potential responders and dissect resistance mechanisms to personalize immunotherapy and combinatorial therapy.

GBM is a genetically heterogeneous disease. Isocitrate Dehydrogenase (IDH)-wildtype GBM tumors can be classified based on genomic, transcriptomic, and DNA methylation data into three main categories, mesenchymal (*MES*), RTKI/proneural (*PN*), and RTKII/RTKIII/classical (*CL*) (*Verhaak et al., 2010*). In addition, other molecular subclasses, such as K27M or G34 mutant have recently been recognized (*Neumann et al., 2016*). *MES* GBM accounts for 30–50% of primary tumors and is associated with particularly poor response to therapy, while *PN* GBM is associated with a somewhat better prognosis. While some reports have shown an enrichment of PD-L1$^{LOW}$ specimens in *PN* GBM and PD-L1$^{HIGH}$ specimens in *MES* GBM (*Berghoff et al., 2015*), PD-L1 tumor expression has not been shown to directly predict clinical outcomes (*Taube et al., 2014*). Molecular GBM subgroups are associated with distinct histological patterns, suggesting that tumor microenvironmental features reflect the specific underlying molecular genetic abnormalities. In addition, GBM contain a highly immunosuppressive tumor microenvironment with abundant tumor-associated macrophages (TAMs), low number of cytotoxic T lymphocytes (CTLs) (*Razavi et al., 2016*; *Nduom et al., 2015*). The role of GBM molecular subtype and impact on tumor immune microenvironment and anti-PD-1 immunotherapy remain poorly understood.

Improving the clinical use of anti-PD-1 immunotherapy in GBM patients requires a comprehensive understanding of tumor genetics and microenvironment as well as the ability to dissect the dynamic interactions among GBM and immune suppressor cells, particularly TAM (*Hambardzumyan et al., 2016*). TAM represents the majority of immune population in GBM (30%–50%), and high TAM density correlates with poor prognosis, and resistance to the therapy (*Hambardzumyan et al., 2016*). We (*Lu-Emerson et al., 2013*; *Cui et al., 2018*) and others (*Thomas et al., 2012*) recently demonstrated that GBM secrete immunosuppressive factors including transforming growth factor-β1 (TGF-β1), and colony-stimulating factor-1 (CSF-1) polarizing monocytes toward an immunosuppressive 'M2-like' phenotype. An in silico analysis of immune cell types in patient GBM biopsies found that the M2-TAM gene signature indicated a greater association with the *MES* subtype (13%) compared to the non-*MES* subtypes: *CL* (6%) and *PN* (5%) (*Wang et al., 2017*). TAM-targeting agents like CSF-1 receptor (CSF-1R) inhibitor have shown promise by reprogramming M2-TAMs toward an anti-tumorigenic 'M1' phenotype in murine glioma models, yet clinical trials on GBM patients showed poor response and patients acquired resistance by the tumor microenvironment (*Pyonteck et al., 2013*). While numerous clinical trials are under way to explore combining anti-CSF1R and immunotherapy (*Cannarile et al., 2017*), there are no biomarkers that could identify patients who could benefit from such combination. A recent failed Phase III immunotherapy clinical trial (CheckMate-498: NCT02617589) (*Hosea, 2019*) emphasizes the need for better identification of patients that may benefit from immunotherapy.

The inability to predict immunotherapy efficacy and identify therapy resistance mechanisms is a major challenge in immuno-oncology including neuro-oncology (*Agrawal et al., 2014*). Discrepancies between preclinical and clinical results have raised concerns about the predictive value of current animal and patient explant culture models and how the findings from the animal models can be translated to patients. While patient-derived xenografts (*Xu et al., 2018*; *Huszthy et al., 2012*) and explant cultures (*Shimizu et al., 2011*) are considered as the gold standard in preclinical validation, there are significant limitations such as lack of accurate humanized immunity and spatiotemporal evolution of GBM tumor niche interactions (*Binnewies et al., 2018*). In vitro bioengineering approaches and tumor-on-a-chip strategies can provide additional high-throughput low-cost avenue to test novel therapies and perform patient screening. A few recent three-dimensional (3D) tissue engineering approaches with microfluidics and 3D bioprinting have been able to model human GBM tumor stromal microenvironments (*Xiao et al., 2019*; *Yi et al., 2019*; *Linkous et al., 2019*), or patient-derived tumor organoids included human immune component (*Moore et al., 2018*). While these methods have a clear advantage for high-throughput and clinical relevant analysis, establishing an orthotopic tumor microenvironment for molecularly distinct GBM subtypes to interrogate the dynamic patient-specific tumor-immune interactions in response to immunotherapy remains a challenge.

Here, we integrated critical hallmarks of the immunosuppressive GBM microenvironments in a microfluidics-based ex vivo microphysiological system termed 'GBM-on-a-Chip'. We aimed to utilize the system to identify potential therapy responses in a cohort of molecularly distinct GBM patients. At a single-cell resolution, we were able to perform longitudinal analysis of allogeneic CD8+ T-cells trafficking through 3D brain microvessels, infiltrating brain-mimicking tissue, and interact with TAMs and GBM cells. By employing cellular (immune cell infiltrate composition, phenotypes, and dynamics), genomic and epigenetic (DNA), transcriptomic (RNA), and proteomic (cytokines) microenvironmental signatures, we dissected the immune-regulatory mechanisms of the GBM microenvironment that evoke resistance to PD-1 inhibition, and showed that co-targeting of PD-1 immune checkpoint and TAM-associated CSF-1R signaling may improve therapeutic efficacy in GBM. Hence, our ex vivo patient-specific 'GBM-on-a-Chip' may significantly lead to personalized immunotherapy screening, improving therapeutic outcomes in GBM patients.

## Results

### Clinicopathological markers fail to predict PD-1 response

To explore the heterogeneity of the immunosuppressive GBM microenvironments, we analyzed a cohort of IDH-wildtype GBM tumors (*Figure 1A*) from patients treated with PD-1 inhibitor (nivolumab) for 2–15 months (median 3.7 months). All primary tumors were classified by clinically validated and New York State approved whole genome DNA methylation classification (*Capper et al., 2018*), MGMT promoter methylation, RNA expression, DNA Copy-Number and next-generation sequencing mutation analysis (*Figure 1—figure supplement 1A*; *Bayin et al., 2016a*). Diagnostic samples were analyzed for PD-L1 and CD163 expressions by immunohistochemistry. Our clinical data indicated that PD-L1 staining was not predictive of response, showing both strong or no expression in both responders and non-responders across different GBM subtypes (*Figure 1B*). Meanwhile, all tumors showed marked TAM infiltration by CD163, irrespective of molecular subtype (*Figure 1B*). The aggressive TAM infiltration present in the perivascular, infiltrative and tumor bulk regions was concurrent with GBM tumor progression, indicating TAM-GBM tumor interactions contribute to the immunosuppressive GBM microenvironments and therapy resistance. GBM methylCIBERSORT analysis of a cohort of 435 glioma samples we previously profiled (*Capper et al., 2018*) further revealed prominent CD14 monocytic and neutrophilic DNA methylation and low CD8+ T-cell methylation signatures in *MES* patients, and low CD14 monocytic and neutrophilic DNA methylation and high CD8 + T-cell methylation in *PN (RTK_I)* patients (*Figure 1C*). However, there was no significant difference between responders and non-responders in DNA methylation in these immune cell signatures (*Figure 1—figure supplement 1*). In addition, our clinicopathological methylation analysis also revealed diverse immunosuppressive signatures in distinct GBM subtypes, and some differences in epigenetic signatures between responders and non-responders (*Figure 1D*), but these were insufficient for predicting the response. These data together confirm that current static biomarkers seem to be poor predictors of anti-PD-1 immunotherapy response, and a further analysis of the heterogeneity of the immunosuppressive GBM microenvironments might help identify niche-associated mechanisms for predicting and improving patient-specific responses to immunotherapy.

### Modeling the GBM tumor niche in an ex vivo 'GBM-on-a-Chip' microphysiological system

To dissect the heterogeneity of anti-PD-1 immunotherapy responses in molecularly distinct GBM cohort of *PN*, *CL*, and *MES* subtypes, we developed a microfluidics-based 3D 'GBM-on-a-Chip' microphysiological system (*Figure 2* and *Figure 2—figure supplement 1*) mimicking the subtype-specific in vivo GBM tumor niche. In this organotypic system, we housed a 3D brain microvessel (*Figure 2A–C*, *yellow*) derived from human brain microvascular endothelial cells (hBMVECs), TAMs derived from human macrophages (*Figure 2—figure supplement 2A and B*), patient-derived and molecularly-distinct GBM cells (*Figure 2A–C*, *red*), and sorted allogeneic human CD8+ T-cells (*Figure 2A–C*, *green*) from primary peripheral blood mononuclear cells (PBMCs) within a 3D brain-mimicking hyaluronan (HA)-rich Matrigel extracellular matrix (ECM) (*Figure 2—figure supplement 2*; *Wang et al., 2019*) (details see Materials and methods). Specifically, 'GBM-on-a-Chip' culture was compartmentalized by a peripheral channel designated for patterning 3D brain microvessels

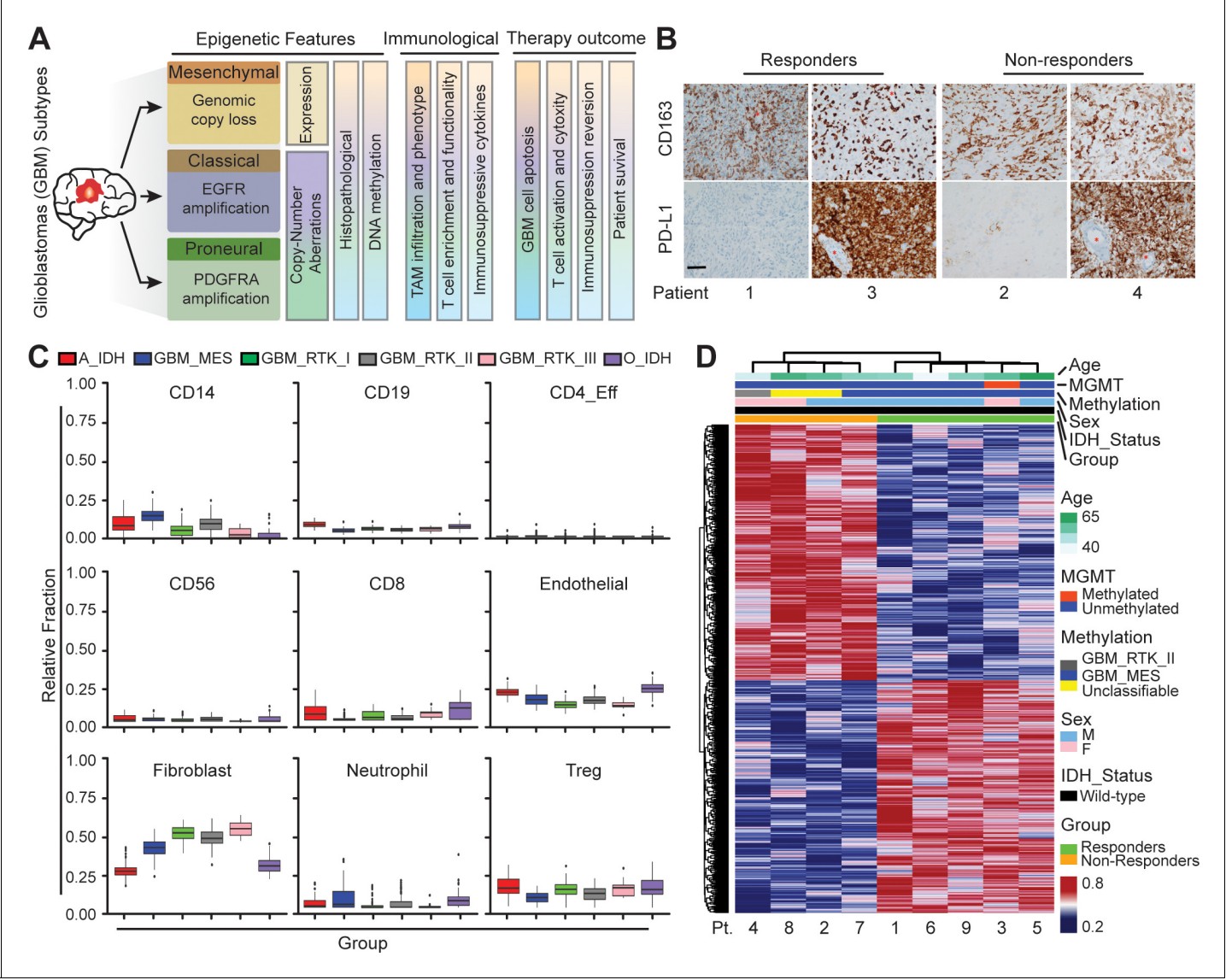

**Figure 1.** Distinct systemic immunosuppression in clinical GBM patients. (**A**) A schematic illustrating the stratification of genetic, molecular and cellular characteristics in distinct GBM subtypes. (**B**) Immunohistochemical analysis of PD-L1 expression on GBM tumors and CD163+ expression on TAM infiltrate. Varied PD-L1 and CD163 expressions with high or no expression in both responding and non-responding GBM patients with administration of a PD-1 inhibitor (nivolumab). Red stars denote brain microvessel. Scale bar is 100 µm. (**C**) MethylCIBERSORT deconvolution of whole genome DNA methylation data from 435 glioma patients (*Capper et al., 2018*) sorted into six main molecular diffuse glioma subtypes (IDH mutated astrocytoma and oligodendroglioma A_IDH and O_IDH; GBM subtypes Mesenchymal: *MES*, Proneural: *RTK_I*, and Classical: *RTK_II* and *RTK_III*) shows variability in immune cell subpopulations across GBM subtypes. p-Values for Kruskal-Wallis test are as follows for CD14 ($p<2.2^{-16}$), CD19 ($p<2.2^{-16}$), CD4_Eff ($p=5.9^{-4}$), CD56 ($p=1.2^{-3}$), CD8 ($p<6.9^{-14}$), Endothelial ($p<2.2^{-16}$), Fibroblast ($p<2.2^{-16}$), Neutrophil ($p=2.1^{-11}$), and Regulatory T-cells (T$_{reg}$) ($p=2^{-15}$). CD14 and CD8 were used to identify the monocytic/macrophage and effector T-cell fractions. (**D**) Clinicopathological information and whole genome DNA methylation showing top 10,000 differentially methylated probes of GBM patients treated with PD-1 inhibitor (nivolumab). Clustering is represented for Responders and non-Responders, irrespective of molecular subtype or other clinicopathological variables (*N* = 9).

The online version of this article includes the following figure supplement(s) for figure 1:

**Figure supplement 1.** Variability in immune cell subpopulations in both responding and non-responding GBM patients with administration of a PD-1 inhibitor (nivolumab).

(outer ring), an intermediate tumor stromal area (middle ring), and a core media region (center region) for long-term media supply (*Figure 2A* and *Figure 2—figure supplement 1*). The three compartments were segregated by regularly spaced micropillars that confine cell-embedded hydrogels to mimic the native in vivo pathological architecture of GBM tumors. To reconstitute in vivo GBM

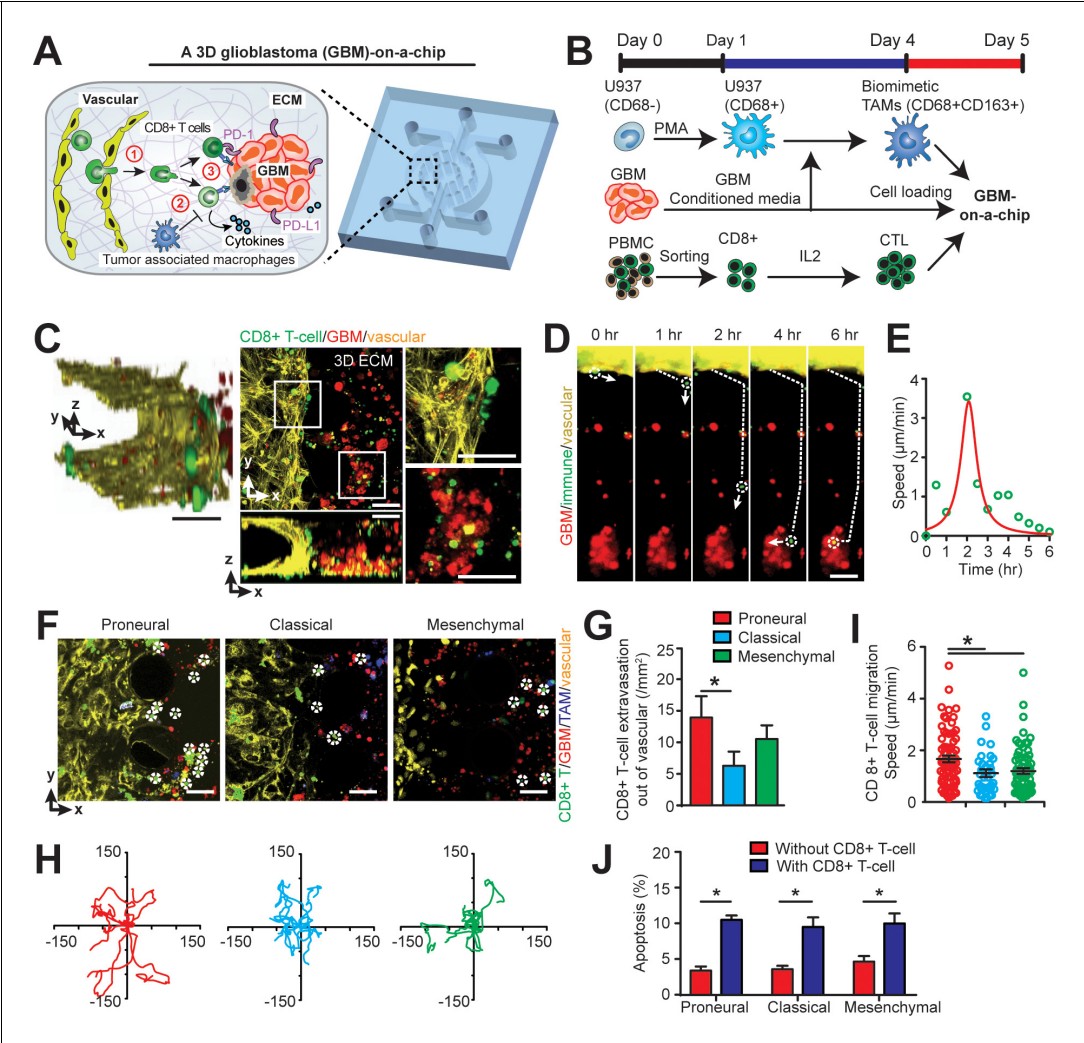

**Figure 2.** Modelling the in vivo GBM tumor niche in a 'GBM-a-on-Chip' microphysiological system. (**A**) A schematic diagram illustrating a microfluidics-based GBM-on-a-Chip model to investigate ① the interactions of immune cell (CD8+ T-cells) with brain microvessels, ② tumor-associated macrophages (TAMs) and ③ GBM tumor cells in an engineered 3D brain-mimicking ECM. (**B**) A schematic illustrating the procedures of cell preparation in the microphysiological system. Biomimetic TAMs (CD68+CD163+) were prepared by differentiating monocyte-like U937 cells with 5 nM of PMA for 24 hr, followed by treatments of conditioned-media of GBM cells for 3 days. Simultaneously, fresh allogeneic CD8+ T-cells were isolated from PBMCs and activated and expanded for 3 days with IL-2. (**C**) Representative confocal immunofluorescence images showing a 3D brain microvessel lumen (*yellow*) in contact with CD8+ T-cells (*green*) and GBM (*PN*, GBML20) tumor cells (*red*). Scale bar is 50 µm. (**D**) Representative time-lapsed images showing a single CD8+ T-cell extravasating through brain microvessels (*yellow*, 0–1 hr), infiltrating through ECM (1–4 hr), and interacting with GBM tumor cells (*red*, 4–6 hr). Scale bar is 50 µm. (**E**) Quantified CD8+ T-cell migration speed at different time points of infiltration, indicating the relatively maximum migration speed after extravasation and before contacting with GBM cells. (**F**) Representative immunofluorescence images showing the distinct counts of allogeneic CD8+ T-cell infiltrate in the *PN* (GBML20), *CL* (GBML08) and *MES* (GBML91) GBM subtypes in GBM-on-a-Chip after 3 days' culture. Note that CD8+ T-cells (*green*) were in contact with brain microvessels (*yellow*), TAMs (*blue*) and GBM tumor cells (*red*). Scale bar is 50 µm. (**G**) Quantified results showing more infiltrated allogeneic CD8+ T-cells in the *PN* GBM as compared to the *CL* and *MES* GBMs. (**H**) Migration trajectories of infiltrated CD8+ T-cell (n > 20) for 2 hr in different GBM subtypes. (**I**) Quantified migration speed of infiltrated CD8+ T-cell, showing faster migration speed in the *PN* GBM as compared to the *CL* and *MES* GBMs at the observation window. Note that the speed range (0–6 µm/min) represents different infiltration stages of different T-cells. (**J**) Quantified GBM cell apoptosis ratio with the presence or absence of IL-2-activated allogeneic CD8+ T-cell in different GBM niches based on caspase-3/7 activation. Error bars represent ± standard error of the mean (s.e.m.). p-Values were calculated using the Student's paired sample *t*-test. *, p<0.05.

The online version of this article includes the following figure supplement(s) for figure 2:

**Figure supplement 1.** Microfabrication of the microfluidics-based 'GBM-on-a-Chip' microphysiological system.

**Figure supplement 2.** Sample preparation for TAMs and effector CD8+ T-cells.

**Figure supplement 3.** CD8+ T-cell extravasation and infiltration behaviors in the engineered GBM microenvironment without the presence of TAM.

**Figure supplement 4.** TAM motility and adherent behaviors in the engineered tumor microenvironments of different GBM subtypes.

tumor composition in vitro, biomimetic human TAMs were nested within the 3D engineered HA-rich ECM tissue (*Figure 2F*, *blue*), making up 30% cell volume (*Hambardzumyan et al., 2016*). TAMs were differentiated from U937 monocytes with Phorbol 12-myristate 13-acetate (PMA) and treated with conditioned media from patient-derived adult GBM cells of all three major molecular subtypes (*Figure 2B*; *Shi et al., 2017*). To mimic in vivo extravasation events of adaptive immune responses of CTLs for different GBM patients, we circulated IL-2-activated allogeneic human CD8+ T-cells into the 3D brain microvessel and profiled their extravasation dynamics as they migrate through brain vasculature, interact with TAMs, and interact with GBM tumor cells (*Figure 2D–J*).

## Distinct extravasation and cytotoxic activities of allogeneic CD8+ T-cells in GBM subtypes

To mechanistically understand CTL activity across molecularly distinct GBMs, we charted in real-time the dynamic extravasation, migration, and cytotoxic activities of primary allogeneic human CD8+ T-cells in the engineered GBM niches. Under time-lapsed imaging (*Figure 2D*), we monitored, on 'GBM-on-a-Chip', a single CD8+ T-cell's extravasation in three stages: transmigration (0–1 hr) through the patterned brain microvessel, penetration (1–4 hr) into the brain-mimicking tissue construct, and interactions with GBM tumors (4–6 hr) at a single-cell level (*Figure 2E*). We quantified the number of allogeneic CD8+ T-cell extravasation in molecularly distinct GBMs (*Figure 2F and G*), cell migration trajectories (*Figure 2H*) and migration speed (*Figure 2I*). We demonstrated that *PN* (GBML20) GBM exhibited significant increases both in the number and speed of allogeneic CD8+ T-cell infiltrate compared to the *CL* (GBML08) and *MES* (GBML91) GBM samples after 3 days' culture (*Figure 2G–I*), which is consistent with our clinical observations (*Figure 1C*). Moreover, we observed stark differences in T-cell migration trajectories, where the *PN* GBM demonstrated free, active motion of CD8+ T-cell while the degree of CD8+ T-cell motility was limited in the *CL* GBM and, arrested in the *MES* GBM (*Figure 2H*). Interestingly, in the absence of TAM, increased number of allogeneic CD8+ T-cell extravasation (15 cell/mm$^2$) was observed, suggesting that the presence of TAM inhibits CD8+ T-cell extravasation (*Figure 2—figure supplement 3*). Our data also showed that the *MES* GBM-educated TAM exhibited faster motion towardsthe brain microvessels, relative to the *PN* GBM-educated TAM (*Figure 2—figure supplement 4*).

The cytotoxic activities of IL-2-activated allogeneic CD8+ T-cell on different subtypes of GBM cells were confirmed with a higher apoptosis ratios of GBM cells as compared to that of without allogeneic CD8+ T-cell in the niche (*Figure 2J*). To further understand the cytotoxic function of the allogeneic CD8+ effector T-cell, we stained the T-cell in different GBM niches with T-cell activation markers CD154 and CD69, and cytotoxic function markers such as granzyme B (GZMB) and perforin (PFN) (*Figure 3*, *Figure 3—figure supplement 1*). The on-chip staining showed that most of IL-2-activated allogeneic CD8+ T-cells expressed GZMB and CD69 but weak PFN after cultured in the GBM niches, while CD8+ T-cell in the *MES* GBM niche overall had lower expressions of these T-cell activation and cytotoxic function markers (*Figure 3—figure supplement 1*), suggesting the immunosuppressive feature of the *MES* GBM niche. Meanwhile, both immunostaining (*Figure 3B*) and qPCR analysis (*Figure 3C*) showed that the ratio of activated CD154+CD8+ T-cells were markedly decreased in all GBM subtypes, while more significantly in the *MES* GBM niche, when compared to CD8+ T-cells cultured without the GBM niche. It confirmed that the immunosuppressive milieu hindered CTL activation and cytotoxic function at different levels of severity in molecularly distinct GBM subtypes.

## GBM subtypes differentially regulate TAM phenotype, epigenetics, and immunity

We next determined the phenotypic status of cytotoxic CD8+ T-cell, TAM, and GBM cell across GBM molecular subtypes (*Figure 3A*). We conducted off-chip cell staining after on-chip cell recovery with specific cell surface markers for CD8+ T-cell activation (CD154+), immune checkpoints (PD-1 and PD-L1) and macrophage phenotype [CD68 for identifying macrophage and CD163 for anti-inflammatory 'M2'-like TAM (*Lu-Emerson et al., 2013*)]. As PD-1 expression is a marker of T-cell activation, we compared and normalized the PD-1 expressions on different GBM-activated allogeneic CD8+ T-cells to the baseline value of PD-1 expression on CD8+ T-cell without tumor activation with immunostaining (*Figure 3B*) and qPCR analysis (*Figure 3C*). Our results confirmed that while PD-1

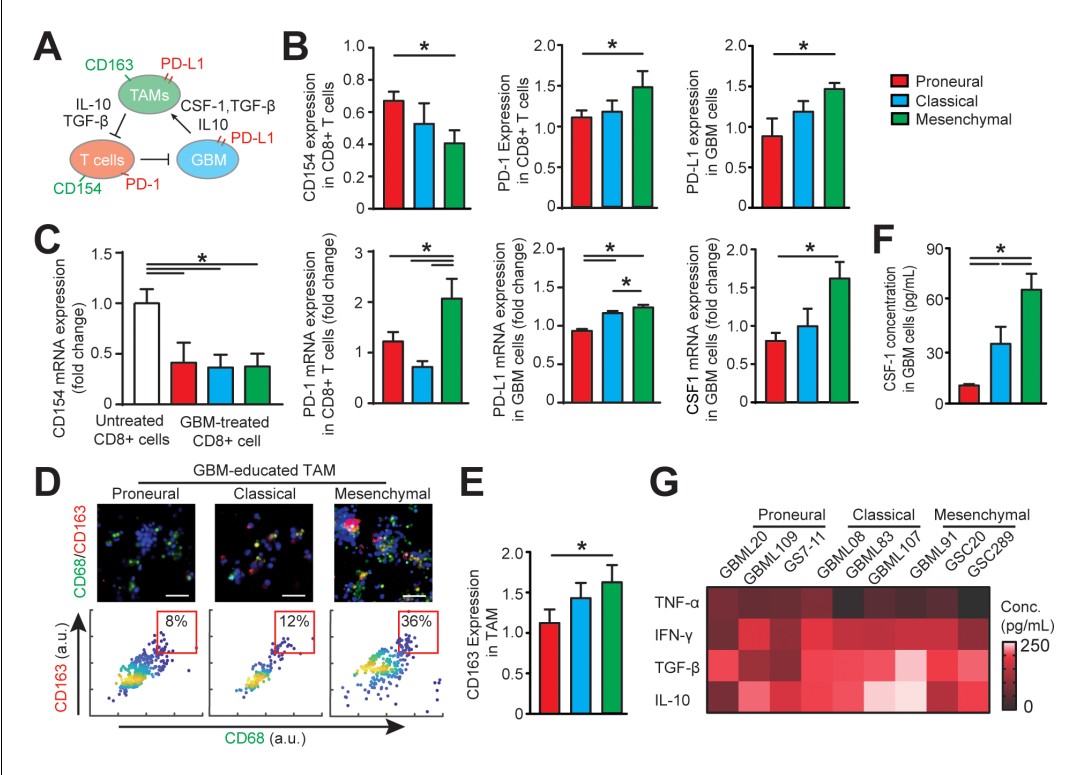

**Figure 3.** Distinct systemic immunosuppression in *PN*, *CL* and *MES* GBMs. (**A**) A schematic highlighting the systemic immunosuppressive signaling among GBM, TAM and CD8+ T-cell *via* CSF-1/CSF-1R, immunosuppressive cytokines and PD-1/PD-L1. (**B**) Quantified CD154 and PD-1 expressions (normalized to untreated) on allogeneic CD8+ T-cell, and PD-L1 expression (normalized to untreated) on GBM cells of *PN* (GBML20), *CL* (GBML08) and *MES* (GBML91) subtypes, showing higher expressions of PD-1 and PD-L1 in the *MES* GBM niche as compared to the *PN* and *CL* GBM niches. Surface marker expression was quantified by the mean intensity of each cell. (**C**) qPCR analysis showing different CD154 and PD-1 expressions in CD8+ T-cell, PD-L1 and CSF-1 expressions in GBM cell. (**D**) Representative immunofluorescence images showing more immunosuppressive M2-like macrophages in the *MES* GBM (GBML91) than the *PN* and *CL* subtypes. Scale bar is 50 µm. (**E**) Quantified M2-like marker CD163 expression (normalized to untreated group) on TAM, in different GBM subtypes, showing higher TAM CD163 expression in *MES* GBM compared to *PN* (GBML20) and *CL* (GBML08) GBMs. (**F**) ELISA results showing high CSF-1 level expressed by *MES* (GBML91) GBM. (**G**) Quantified cytokine levels in different GBM derived niches, showing relatively higher expressions of immunosuppressive cytokine (TGF-β1 and IL-10), lower expressions of pro-inflammatory cytokines (IFN-γ and TNF-α) in *MES* GBM (GBML91). Error bars represent ± s.e.m., n > 80 in B, D, and F. *P*-values were calculated using one-way ANOVA. *, p<0.05.

The online version of this article includes the following figure supplement(s) for figure 3:

**Figure supplement 1.** Analysis of allogeneic CD8+ T-cell activation in various GBM niches.

**Figure supplement 2.** Comparison of cellular and cytokine conditions of IDH-mutant and IDH-wildtype GBM tumor cells.

expression was low on untreated PBMC-derived CD8+ T-cell, it was elevated on the tumor-activated PBMC-derived CD8+ T-cells in most GBM niches. Particularly, the *MES* GBM tumor niche was characterized with highest expressions of PD-1 on CD8+ T-cell and PD-L1 on patient-derived GBM cell (*Figure 3B*). Further qPCR analysis confirmed the strong levels of PD-1 expression in *MES* GBM treated CD8+ T-cell and PD-L1 in *MES* GBM cell (*Figure 3D*). The PD-L1 mRNA expressions in GBM cell (*Figure 3C*) and TAM (*Figure 5—figure supplement 1*) varied across different GBM subtypes. Since the majority of intra-tumoral immune cells in GBM were represented by TAM (*Figure 1B and C*), we analyzed the immunosuppressive TAM activity in GBM-on-a-Chip. Both immunostaining (*Figure 3D–E*) and qPCR analysis (Figure 5C) results showed a significant number of immunosuppressive CD163+ M2-TAMs in the *MES* GBM compared to that in the *PN* and *CL* subtypes, which is consistent with our patient sample immunohistochemistry and methylCIBERSORT data (*Figure 1*).

To further analyze the properties of the immunosuppressive cytokine milieus across GBM patient subtypes, we mapped the anti- and pro-inflammatory cytokines by using enzyme-linked immunosorbent assay (ELISA) in nine patient-derived GBM lines. CSF-1 has been shown to influence

macrophage polarization toward a M2 phenotype in GBM (*Pyonteck et al., 2013*); however, it is unclear if molecularly different GBM subtypes have distinct CSF-1 secretion profiles. We demonstrated here that CSF-1 was highly secreted in the *MES* GBM, compared to the *PN* and *CL* subtypes using qPCR analysis (*Figure 3C*) and ELISA assay (*Figure 3F*). These results thus suggest CSF-1 signaling as an ideal therapeutic target in all GBM subtypes (*Zhu et al., 2014*). Furthermore, our results showed that different GBM patient-derived cells showed distinct immunosuppressive cytokine milieus, and *MES* and *CL* GBMs likely had higher productions of anti-inflammatory cytokines TGF-β1 and IL-10, compared to the pro-inflammatory cytokines TNF-α and IFN-γ (*Figure 3G*), driving TAM polarization toward a M2-like phenotype. IDH-wildtype GBMs have been reported to display a greater number of tumor-infiltrating lymphocytes and elevated PD-L1 expression compared to IDH-mutant GBMs (*Berghoff et al., 2017*), thus IDH mutational status may contribute differently to adaptive immune responses. However, the IDH-mutant GBM (patient MGG152) showed similar cellular and cytokine conditions to the IDH-wildtype *PN* GBM in our on-chip study, which may be due to the poor survival capability of these IDH-mutant GBM cells when cultured in vitro (*Figure 3—figure supplement 2*).

To assess the epigenetic modifications of TAM in molecularly distinct GBMs (*CL*, *PN*, and *MES*), we analyzed the DNA methylation of TAM by recovering macrophages and GBM cells from the GBM-on-a-Chip culture and performing whole genome DNA methylation analysis. We found that culturing tumor cells with the presence or absence of macrophages in the niche resulted in different epigenetic profiles and *vice versa,* the presence of GBM cell also altered the DNA methylation signatures of co-cultured macrophages (*Figure 4*). Rap1 signaling pathway, a known regulator of T-cell and antigen-presenting cells (*Katagiri et al., 2002*), was upregulated both in co-cultured GBM and macrophage cells, when compared to the GBM cells and macrophages in mono-cultures (*Figure 4—figure supplement 1*). Interestingly, PD-L1 promoter methylation was slightly hypomethylated in mono- and co-cultured GBM cells (*Figure 4—figure supplement 2*). Combined our DNA methylation results suggest that the interaction between TAM and GBM might regulate cytotoxic CD8+ T-cell activation *via* Rap1 signaling particularly in the *CL* GBM subtype. Also, PD-L1 expression was not epigenetically silenced in the absence of TAM. In addition, we examined the dynamic interactions between the GBM and ECM over 3 days (*Figure 4—figure supplement 3*) but did not found significant changes in the deposition of HA, laminin, collagen IV and fibronectin in different GBM subtypes.

## Optimizing anti-PD-1 therapy by co-targeting TAM CSF-1 signaling

Despite early reports of response to immunotherapy, a recent Phase 3 CheckMate-498 study using PD-1 blockade nivolumab in MGMT–unmethylated newly diagnosed GBM failed to meet primary endpoints (*Hosea, 2019*) highlighting the need to better stratify patients and identify potential responders as well as testing potential combinational therapies. Using our patient-specific GBM-on-a-chip system, we tested an adjuvant strategy that simultaneously targeted M2-TAM polarization and PD-1 immune checkpoint with nine GBM patient-derived molecularly distinct cell lines (*Figure 5A*). To screen GBM subtype-specific responses, we delivered monotherapy or dual-therapy regimens of brain-penetrant CSF-1R inhibitor (BLZ945, 0.1 μg/ml) to ablate TAM immunosuppressive function and human IgG4 anti-PD-1 monoclonal antibody (nivolumab, 1 μg/ml) to inhibit the PD-1/PD-L1 pathway every 24 hr for 3 days. Consistent with previous studies (*Pyonteck et al., 2013*; *Zhu et al., 2014*), BLZ945 suppressed the polarization of macrophages toward an immunosuppressive M2 phenotype in all three GBM subtypes with more significant CD163 marker suppression in the *CL* and *MES* GBM subtypes relative to the *PN* GBM (*Figure 5B and C*). However, BLZ945 treatment alone caused no significant change in PD-L1 expression for both TAM and GBM cells (*Figure 5—figure supplement 1*), implying that BLZ945 monotherapy cannot completely abolish the systemic immunosuppression in the GBM microenvironments.

We examined allogeneic CD8+ T-cell extravasation in the different molecular subtypes of GBM tumors under control (vehicle), BLZ945 or nivolumab monotherapy, and 'dual' BLZ945 and nivolumab therapy regimens. Our results indicated that CSF-1R blockade can significantly enhance allogeneic CD8+ T-cell extravasation across brain microvessels, compared to PD-1 blockade alone (*Figure 5D*). However, as demonstrated by profiling CD154 expression, targeting TAM with BLZ945 alone did not significantly reverse the immunosuppression onto the cytotoxic CD8+ T-cell, compared to the untreated condition (*Figure 5E*). In addition, PD-1 inhibition alone did not increase the

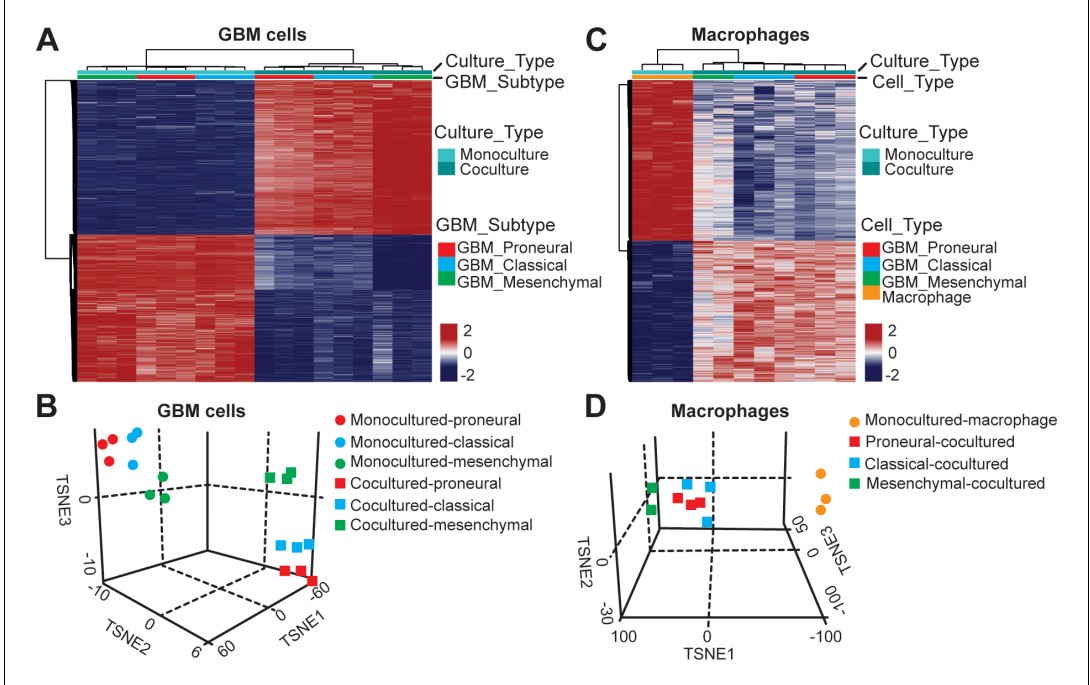

**Figure 4.** DNA methylation analysis of interactions between patient-derived GBM cell and macrophage in an engineered 3D GBM niche environment. (A) Whole genome DNA methylation analysis showing top 10,000 differentially methylated probes of patient-derived *PN* (GBML20), *CL* (GBML08) and *MES* (GBML91) GBM cells cultured in a 3D brain-mimicking ECM environment with or without macrophages. (B) tSNE analysis of mono-cultured and co-cultured GBM cells showing clear separation of all molecular GBM subtypes, *PN* (GBML20), *CL* (GBML08) and *MES* (GBML91) (each in triplicate) in the same direction when exposed to macrophages. However, the effect appears to be different in the three molecular subtypes with *PN* GBM mostly affected by presence of macrophage. (C) Whole genome DNA methylation analysis showing top 10,000 differentially methylated probes of mono-cultured and GBM cell-educated macrophages. (D) tSNE analysis of mono-cultured and patient-derived GBM cell-educated macrophages showing distinct shifts in methylation in all molecular GBM subtypes. However, *MES* GBM cell co-cultured macrophages cluster showed a more distinct separation.

The online version of this article includes the following figure supplement(s) for figure 4:

**Figure supplement 1.** Top KEGG pathways between mono-cultured and co-cultured GBM cells in a 3D brain-mimicking ECM environment.
**Figure supplement 2.** PD-L1 promoter methylation in mono-cultured and co-cultured GBM cells in a 3D brain-mimicking ECM environment.
**Figure supplement 3.** Analysis of extracellular matrix composition in different engineered GBM niches.

extravasation of allogeneic CD8+ T-cell in GBM-on-a-Chip but did enhance CD8+ T-cell activation in the *PN* and *MES* GBMs. PD-1 and CSF-1R dual blockade increased the extravasation of allogeneic CD8+ T-cell across brain microvessels (*Figure 5D&E*), reversed the immunosuppression onto CD8+ T-cell in terms of TNF-α and TGF-β1 production (*Figure 5F*, *Figure 5—figure supplement 2*), and augmented CD8+ T-cell cytotoxic function with higher GBM tumor apoptosis shown by caspase-3/7 activation (*Figure 5G–H*, *Figure 5—figure supplement 3*) for each GBM subtype, specifically *the MES* GBM, compared to monotherapies.

In the brain microenvironment, microglia are the brain-resident macrophages and can play a similar role or cooperate with blood-borne macrophages to regulate brain tumor development and therapy response. Our results showed that the presence of microglia in the GBM microenvironment could promote the PD-1 expression on allogeneic CD8+ T-cell (*Figure 5—figure supplement 4A*), but there was no significant change observed in GBM cell apoptosis response to the PD-1 and CSF-1R dual blockade treatment compared to the macrophage only system (*Figure 5—figure supplement 4B&C*). Altogether, our pre-clinical screening in the biomimetic GBM-on-a-Chip demonstrated that co-targeting M2-TAM could serve as a potential combinational therapy strategy for improving anti-PD-1 immunotherapy.

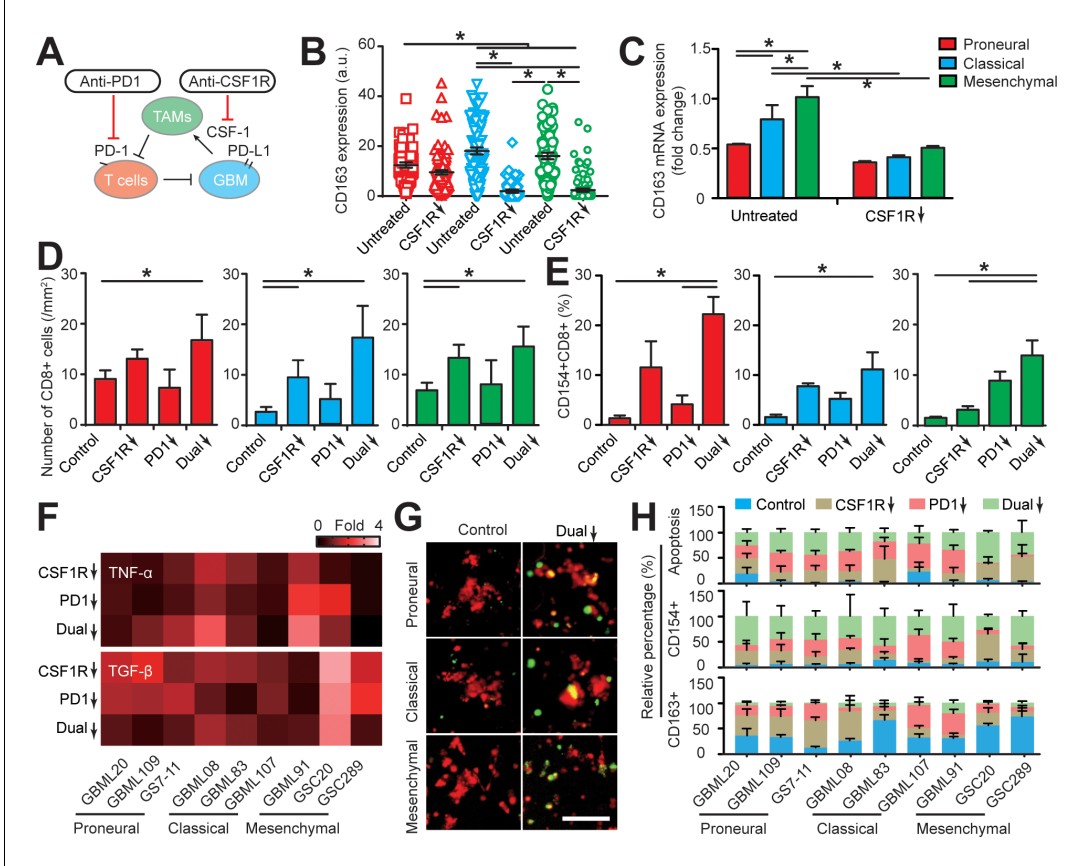

**Figure 5.** Targeting TAM with anti-CSF-1R blockade improves anti-PD-1 immunotherapy response in GBM-on-a-Chip. (**A**) A schematic outlining a dual inhibition therapeutic strategy for targeting both PD-1/PD-L1 and TAM CSF/CSF-1R signaling to inhibit the systemic immunosuppression among GBM, TAM and CD8+ T-cell. (**B**) Quantified M2-like marker CD163 expression on TAM in response to CSF-1R inhibitor BLZ945 in different GBM subtypes (GBML20, GBML08, and GBML91), showing the limited expression of CD163 in all GBM subtypes. (**C**) qPCR experiment confirming the inhibition of CD163 expression in TAM with the administration of CSF-1R inhibitor BLZ945. (**D**) Quantified results showing more infiltrated allogeneic CD8+ T-cells in all GBM subtypes (GBML20, GBML08, and GBML91) with PD-1 and CSF-1R dual inhibition therapy as compared to Nivolumab and BLZ945 monotherapy. (**E**) Quantified results showing an increased influx of activated CD154+CD8+ T-cells in PD-1 and CSF-1R dual inhibition therapy as compared to Nivolumab monotherapy. (**F**) Quantified cytokine levels showing significantly increased expression of pro-inflammatory cytokine (TNF-α) and decreased expression of immunosuppressive cytokine (TGF-β1) in most GBM subtypes with dual inhibition therapy. Fold changes were calculated relative to control. Note the patient-specific responses with different pharmacological treatment. (**G**) Representative apoptosis images showing more apoptotic (green nuclei) GBM cells with co-blockade of PD-1 and CSF-1R relative to control in all GBM subtypes (GBML20, GBML08, and GBML91). Live GBM cells were stained with CellTracker Red (red color). (**H**) A therapeutic response summary of the heterogeneous and systemic immunosuppression in nine lines of GBM patients' derived cells using GBM-on-a-Chip for relative percentages of GBM cell apotosis, CD154+CD8+ (%) and CD163+CD68+ (%) cell populations. 100% stacked bar chart was used to show the relative difference among distinct drug treatments. CSF-1R inhibitor BLZ945 (0.1 μg/ml) and PD-1 blockade nivolumab (1 μg/ml) were used in all the monotherapy or dual inhibition treatments. All control groups were treated with fresh cell culture media supplemented with DMSO (0.01%) and human IgG4 isotype control antibody (1 μg/mL, BioLegend). Error bars represent ± s.e.m. from three independent experiments. n > 80 in (**B**), (**D**), (**E**), and (**H**). *P*-values were calculated using the Student's paired sample *t*-test or one-way ANOVA. *, p<0.05.

The online version of this article includes the following figure supplement(s) for figure 5:

**Figure supplement 1.** qPCR analysis showing different immunosuppression in TAM and GBM cell.

**Figure supplement 2.** Cytokine conditions in different patient-derived GBM cell constructed microenvironments.

**Figure supplement 3.** Apoptosis ratios of GBM cells under different drug treatments.

**Figure supplement 4.** Microglia affect CD8+ T-cell PD-1 expression and GBM cell apoptosis.

## Discussion

Rapid progression, a lack of robust clinical biomarkers, and an insufficient clinical response present major challenges for adapting PD-1 checkpoint-based immunotherapy for GBM patients. GBM patients are largely stratified into clinical trials based on MGMT promoter methylation and IDH1/2

mutation status. However, this does not sufficiently reflect the significant inter- and intra-tumoral heterogeneity. A diversity of genetic and immune signatures of patients in response to PD-1 immunotherapy have been reported, but many of these only integrated genomic and transcriptomic readouts of GBM tumors (*Venteicher et al., 2017*) and nivolumab treatment (*Schalper et al., 2019*; *Riaz et al., 2017*). Our clinical and experimental data demonstrated that GBM patients of transcriptionally defined subtypes have distinct epigenetic and immune signatures that may lead to different immunosuppressive mechanisms. Nevertheless, these integrated genetic analyses and cell markers from patient biopsies cannot fully capture the dynamic evolution of the tumor microenvironment in response to therapy (*Riaz et al., 2017*), which may partly account for the limited clinical success of anti-PD-1 therapy.

In the current study, we addressed clinical needs by engineering a human 'GBM-on-a-Chip' microphysiological system to dissect the heterogeneous immunosuppressive GBM microenvironments with a real-time and longitudinal analysis of immune activities under different therapy strategies. Our work differs substantially from previous methods (*Jenkins et al., 2018*; *Neal et al., 2018*), in the aspect of engineering a humanized ex vivo model using patient-derived cells and the capability of real-time monitor of tumor-immune-vascular interactions and therapy responses for screening optimized PD-1 blockade. Importantly, our GBM-on-a-Chip allows for a multidimensional readout of patient-specific responses to different immunotherapy regimens ex vivo on the basis of cellular (immune cell infiltrate composition, phenotypes, and dynamics), epigenetic, transcriptomic, and secretomic signatures to examine the prognostic relationship between patient response and the GBM subtypes (*PN*, *MES*, and *CL*) and genetic mutations (IDH). Our results revealed that different subtypes of GBMs illustrated distinct CD8+ T-cell kinetics as allogeneic CD8+ T-cell extravasate across the 3D brain microvessel, traverse through brain-mimicking tissue, and interact with TAM and patient-derived GBM cells. Using our GBM-on-a-Chip model, we demonstrated that M2-like CD68 +CD163+ TAMs dominated the immunosuppressive microenvironment in the *MES* GBM, by restricting the dynamics of CD8+ T-cell recruitment and activation, which can be effectively reversed with CSF-1R and PD-1 dual blockade therapy. Moreover, targeting immunosuppressive TAM alone with a CSF-1R inhibitor increased allogeneic CD8+ T-cellinfiltration in the tumor, however alone it still yielded a limited effect on tumor apoptosis consistent with previous studies (*Quail et al., 2016*). Similarly, targeting PD-1 alone resulted in a modest effect on allogeneic CD8+ T-cell extravasation. Thus, our findings provide a rationale to combine CSF-1R blockade to optimize the therapeutic effect of immune checkpoint blockade, particularly for the aggressive *MES* GBM. In the brain microenvironment, brain resident microglia are considered as another source of TAM to regulate brain tumor development and therapy response. Our results indicated that microglia in the GBM microenvironment might have a similar immunosuppressive effect with the PBMC-derived macrophages on CD8+ T-cell PD-1 expression and functionality. Yet, no significant change in GBM cell apoptosis response to the PD-1 and CSF-1R dual blockade treatment was observed in our study, which might be contributed by the complex reprogramming of microglia phenotypes in the brain tumor microenvironment as shown previously (*Cannarile et al., 2017*).

The current model might be further improved to replicate a truly patient-specific ex vivo GBM model. First, the allogeneic immune and stromal cells used in the current proof-of-concept GBM model may limit the clinical significance of the findings for patient-specific immunotherapy screening. An autologous model constructed with all patient-derived cells will envision a truly personalized GBM-on-a-Chip to further improve the predictive value of the system. Secondly, intact blood-brain barrier (BBB) can hinder the therapeutically effective drug delivery and limit the drug efficacy in some brain tumors. However, it is well-established that BBB is uniformly disrupted in GBM which leads to leaky blood vessels (*Sarkaria et al., 2018*). Thus, our simplified GBM microenvironment model without BBB construction, although might not perfectly mimic the in vivo structure, can still serve as a suitable and useful model to dissect the GBM tumor-immune-vascular interactions ex vivo. In addition, HA-enriched Matrigel ECM was used in the current model, while the effects of heterogeneous composition of the ECM such as collagen, laminin and HA on cell growth needed to be studied and optimized. We examined the dynamic interactions between the GBM and ECM over 3 days but did not found significant difference in the deposition of HA, laminin, collagen IV and fibronectin in GBM subtypes. It might be because a short-term culture and predefined ECM composition might not be able to reflect the actual in vivo dynamic interactions of GBM and ECM. Taken together, further populating models with autologous patient-derived cells, biomimetic BBB functions, and

tunable ECM-tumor-immune interactions will provide a better system to improve the clinical significance of patient-specific study.

Altogether, we demonstrated the feasibility of a patient-specific screening for immunotherapy responses ex vivo with our GBM-on-a-Chip platform to dissect the heterogeneous tumor immune microenvironments, rationalize and screen effective therapeutic combinations and facilitate precision immuno-oncology. We envision that a truly personalized GBM-on-a-Chip system can significantly accelerate the pace for identifying novel therapeutic biomarkers, developing patient-specific immunotherapeutic strategies, and optimizing therapeutic effect and long-term management for a broader GBM patient population.

## Materials and methods

### Patients

All patients were treated at NYU Langone Health between 6/1/2017 and 3/1/2019. All tumor biopsies were molecular profiled using clinically validated next-generation sequencing, MGMT promoter methylation analysis by pyrosequencing and molecularly classified by clinically validated whole genome DNA methylation as described previously (Capper et al., 2018). Patients received nivolumab 'off-label' for newly diagnosed or recurrent glioblastoma. Nivolumab was administered at 3 mg/kg intravenous injection every 14 days. Median duration of nivolumab therapy was 3.5 months (range 0.5 to 15 months). Patients were assessed clinically once per months and had follow-up MRI assessments every 2 months. Patients were classified as 'responders' if they appeared clinically (improving or stable neurological deficits without need for steroids) and radiographically (MRI demonstrating <25% increase in abnormal enhancement compared to pre-nivolumab baseline MRI brain) for at least 3 months after beginning immunotherapy. Patients were classified as 'non-responders' if they were clinically deteriorating (worsening symptoms, increasing steroid requirements) or if MRI demonstrates $\geq$25% increase in contrast-enhancing within 3 months from start of nivolumab therapy.

### Patient-derived tumor cells and culture

Fresh tumor tissues were harvested from GBM patients undergoing resection surgery of GBM after informed consent (IRB no.12–01130) (Bayin et al., 2016b), and characterized for different GBM subtypes (Supplementary file 1A). Patient-derived cells of GBML08, GBML20, GBML83, GBML91, GBML107, GBML109 (provided by Dimitris G. Placantonakis's lab at New York University School of Medicine) and MGG152 (provided by Andrew S. Chi's lab at New York University Langone's Brain Tumor Center) were cultured in GBM basal medium supplemented with every 2–3 days with 20 ng/ml Epidermal growth factor (EGF, Sigma-Aldrich) and 20 ng/ml basic fibroblast growth factor (bFGF, Sigma-Aldrich). GBM basal medium was prepared with Neurobasal media (21103049, Gibco), 1 × Non Essential Amino Acids (11140–050, Gibco), 1 × B27 (without Vitamin A) (12587–010, Gibco), and 1 × N2 (17502–048, Gibco). Patient-derived cells of GS7-11, GSC20, and GSC289 (provided by Erik P. Sulman's lab at New York University Langone's Brain Tumor Center) were cultured in Dulbecco's modified Eagle's medium (DMEM, Sigma-Aldrich) supplemented with 1 × B27 Supplement (17504–044, Gibco), 20 ng/mL EGF (E9644, Sigma-Aldrich), 20 ng/mL bFGF (F0291, Sigma-Aldrich) and 1% penicillin/streptomycin (Gibco). Parental tumors and cultures derived from them were always profiled with DNA methylation arrays and with RNA-sequencing, to ensure maintenance of the molecular subtype.

### Cell culture and reagents

HBMVECs (10HU-051, iXCells Biotechnologies) were cultured in recommended Endothelial Cell Growth Medium (MD-0010, iXCells Biotechnologies). Cells are collected with 0.05% trypsin-EDTA and subcultured with a plating density of $5 \times 10^3$ cells/cm$^2$. Only early passages of HBMVECs (passage 1–6) are used in our assays. U937 monocytes (ATCC) were maintained in RPMI-1640 medium (Gibco) supplemented with 10% fetal bovine serum (FBS, Gibco) and 1% penicillin/streptomycin (Gibco). Human microglia cell line HMC3 (CRL-3304, ATCC) was cultured in Eagle's Minimum Essential Medium (EMEM, 30–2003, ATCC), supplemented with 10% FBS (Gibco) and 1% penicillin/

streptomycin (Gibco). All the cells were cultured in a 37°C incubator with 5% $CO_2$. These cell lines have been authenticated with the short tandem repeats (STR) profiling and mycoplasma testing.

## TAM and CD8+ T-cell preparation

U937 monocytes were polarized into macrophages with treatments of 5 nM PMA (Sigma-Aldrich) for 24 hr (*Shi et al., 2017*). Biomimetic TAMs (CD68+CD163+) were obtained by culturing U937-derived macrophages in complete culture media supplemented with supernatants of patient-derived GBM cells ($5 \times 10^5$ cells/mL) which were collected after 3 days' culture and centrifuged at $2000 \times g$ for 10 min at 4°C to remove cell debris. Cryopreserved human PBMCs (10HU-003, iXCells Biotechnologies) were thawed and cultured in RPMI-1640 medium (Gibco) supplemented with 10% FBS (Gibco) and 1% penicillin/streptomycin (Gibco) overnight before sorting for CD8+ T-cells. Allogeneic CD8+ T-cells were isolated from PBMCs via negative selection using MojoSort Human CD8 T-Cell Isolation Kit (MojoSort, 480011, Biolegend) as per the manufacturer's protocol (*Figure 2—figure supplement 2*). Isolated CD8+ T-cells were activated and expanded for 2–3 days in PBMC culture medium supplemented with 10 ng/mL recombinant IL-2 (589104, Biolegend).

## Microfluidic chip fabrication

A microfluidic chip containing a set of vascular-seeding channel, hydrogel loading channel, and media infusion channel was used to build the GBM microenvironment. The microfluidic chips were fabricated using the standard soft lithographic method. Briefly, silicon master molds were first fabricated by standard photolithography using SU-8 photoresist (SU8-2075, Microchem) with a thickness of 100 μm. After coating trichloro (1H, 1H, 2H, 2H-perfluorooctyl) silane (448931, Sigma-Aldrich) vapor overnight in vacuum desiccation to facilitate the Polydimethyl siloxane (PDMS, Sylgard 184, Dow Corning) release, PDMS prepolymer was mixed with a curing agent at a weight ratio of 10:1, poured onto the master molds, degassed in a vacuum desiccation for 2 hr to remove air bubbles, and cured at 80°C overnight. Silicone slabs were then cut out from the master molds and punched to make inlets and outlets for vascular-seeding channels, hydrogel loading channels and media infusion channels. Finally, oxygen plasma (350W, PlasmaEtch) was applied to irreversibly bond PDMS slabs and glass coverslips, and then baked overnight at 80°C.

## Synthesis and preparation of brain tissue-mimicking hydrogel

Brain tissue-mimicking hydrogel was prepared by interpenetrating growth-factor-reduced Matrigel matrix (Corning) and matrix metalloproteinase (MMP)-sensitive hyaluronic acid (HA) hydrogels with a volume ratio of 1:1. MMP-sensitive HA hydrogel was synthesized as described previously (*Wang et al., 2019*). Briefly, HA-ADH was firstly obtained using hyaluronic acid (100 mg, 0.0015 mmole, 50 kDa), ADH (2.6 g, 8.4 mmole) and 1-ethyl-3-[3-dimethylaminopropyl] carbodiimide hydrochloride (EDC) (0.3 g, 0.92 mmole) at pH 4.75, followed by dialysis (MWCO 6–8 kDa) in deionized water for 2–3 days and lyophilizing. Acrylated hyaluronic acid (HA-AC) was prepared by reacting the synthesized HA-ADH (100 mg, 0.0014 mmole) with N-Acryloxysuccinimide (NHS-AC) (108 mg, 0.47 mmole) in HEPES buffer overnight and dialysis in DI water for 2–3 days before lyophilizing. HA-AC was further conjugated with RGD peptides (Ac-GRGDSPCG-NH2, Genscript) overnight, dialysis in DI water for 2 days and lyophilizing. Finally, MMP-sensitive HA hydrogel (10 mg/mL) was obtained by crosslinking with MMP-degradable crosslinker (GCRDVPMSMRGGDRCG, Genscript).

## Generation of ex vivo tumor microenvironment

To firmly bond the brain tissue-mimicking hydrogels in the microfluidic chip, fabricated chip was firstly treated with oxygen plasma (350W, 2 min), then coated with 1 mg/mL Poly-D-Lysine (A-003-E, Millipore Sigma) for 2 hr at room temperature. After washing completely with distilled water at least twice, the microfluidic chip was further baked at 80°C overnight to recover the hydrophobic property of the PDMS channels. The microfluidic chip was then transferred into a cell culture biosafety hood, and subsequently sterilized with UV for 30 min. To avoid contamination, all the following procedures were conducted in the sterile microenvironment.

Patient-derived GBM cells were firstly dissociated with Accutase (Innovative Cell Technologies), and then labeled with CellTracker Red (5 μM, C34552, Thermo Fisher-Scientific) as per the manufacturer's instructions. TAMs and GBM cells were then mixed with a number ratio of 1:2 at a cell density

of $1 \times 10^8$ cells/mL in the brain tissue-mimicking hydrogel. The final cell numbers of GBM and TAMs were about $1 \times 10^5$ cells per chip. Notably, cells were first suspended in a HA hydrogel, then loaded into Matrigel on ice to avoid the gelation of Matrigel. Hydrogel solution was then quickly flushed into the hydrogel loading channel, and then incubated in an incubator at 37°C and 5% $CO_2$ for 30 min for complete gelation. After incubation, the media infusion channel and the vascular-seeding channel was flushed with fresh cell culture media for 2 hr. HBMVECs labeled with DiD (5 μM, V22887, Thermo Fisher-Scientific) were seeded into the vascular-seeding channel at a density of $5 \times 10^6$ cells/mL, and incubated at 37°C and 5% $CO_2$ overnight with fresh cell culture media. To form the vascular lumens, the microfluidic chip was flipped every 15 mins for 2 hr to let HBMVECs to uniformly attach to the vascular-seeding channel. After 24 hr' incubation, IL-2activated allogeneic CD8+ T-cells were firstly labeled with CellTracker Green (5 μM, C2925, Thermo Fisher-Scientific) and loaded into the vascular channel at a total cell number of $1 \times 10^6$ cells per chip. After culture in the incubator for 1–3 days, live cell images of different cell compartments were captured by an inverted fluorescent microscope (Zeiss Axio Observer.Z1).

## Cell fixation and immunostaining

To characterize the expression of surface marks on GBM cell, CD8+ T-cell and TAM, cells were firstly recovered with Corning Cell Recovery Solution (Corning, NY, USA) from the brain tissue-mimicking hydrogel. Briefly, cell culture medium was aspirated, and the microfluidics chip was rinsed with cold $1 \times$ PBS twice, 10 min per round. PDMS slabs were detached from the cover glass in the microfluidics chip, followed by incubation with Corning Cell Recovery Solution on ice for 30 min. With gently pipetting the channel containing cells, the cell suspension was centrifuged, washed with cold $1 \times$ PBS twice and then resuspended in 5% BSA blocking buffer. All cell samples were fixed in 4% paraformaldehyde (PFA) (Sigma-Aldrich) for 10 min at room temperature. To identify different surface makers on T-cell, TAM, and GBM cell, cells were first stained with specific primary antibodies (details in *Supplementary file 1B*), and then visualized with secondary antibodies. Specifically, CD8 and CD154 expressions on T-cell were stained with Alexa Fluor 647 conjugated anti-human CD8 (344726, Biolegend) and PE conjugated anti-human CD154 (310805, Biolegend) at 4°C for 30 min. To quantify PD-1 expression on CD8+ T-cell, cells were first incubated with PD-1 primary antibody (367402, Biolegend) for 1 hr, and then visualized with Alexa Fluor 488 conjugated goat anti-mouse IgG secondary antibodies (Invitrogen, 5 μg/mL). TAMs were identified by staining with Alexa Fluor 647 conjugated anti-human CD68 antibody (333819, Biolegend), followed by staining with anti-human CD163 primary (333602, Biolegend) and Alexa Fluor 488 conjugated goat anti-mouse IgG secondary antibodies (Invitrogen, 5 μg/mL), or PD-L1 primary antibody (MAB1561, R and D Systems) and Alexa Fluor 555 conjugated goat anti-mouse IgG secondary antibodies (Invitrogen, 5 μg/mL). To identify PD-L1 expression on GBM cell, GBM cells were labeled with CellTracker Red (5 μM, C34552, Thermo Fisher-Scientific) before loading into the chip. After recovering cells from the chip, PD-L1 primary antibody and Alexa Fluor 488 conjugated goat anti-mouse IgG secondary antibodies (Invitrogen, 5 μg/mL) were used to stain PD-L1 expression. Fluorescent images of stained cell samples were obtained by an inverted fluorescent microscope (Zeiss Axio Observer.Z1). PD-1 and CD154 expressions on CD8+ T-cell, PD-L1 and CD163 expressions on macrophage were quantified based on the mean florescent intensity of cell staining using ImageJ (NIH). Alternatively, ratios of CD154+ or CD163+-cells were calculated relative to CD8+ cells or CD68+ cells, respectively.

## Quantification of cell migration

Infiltrated CD8+ T-cells were defined as the CD8+ T-cells migrating out of vascular lumens. To quantify the migration behaviors of those CellTracker Green-labeled allogeneic CD8+ T-cells, time-lapsed image stacks were acquired every minute for 2 hr and at least three positions in each microfluidic chip. The cell centroids at different time points of the same cell were then used to calculate the cell migration speed and linked up to represent the migration trajectories by using ImageJ (NIH). The mean migration speeds were then averaged for all infiltrated cells.

## CD8+ T-cell activation analysis

To characterize the effector function of allogeneic CD8+ T-cell, cells in the devices were fixed and permeabilized, followed by stained with PE conjugated anti-human CD154 (310805, Biolegend),

FITC conjugated anti-human CD69 (310904, Biolegend), PE conjugated anti-human Perforin Antibody (308106, Biolegend), or PE conjugated anti-human/mouse Granzyme B Recombinant Antibody (372208, Biolegend). Fluorescent images were obtained by a fluorescent microscope (Zeiss Axio Observer.Z1) with a 40 × objective, then analyzed using ImageJ (NIH).

## Quantification of GBM cell apoptosis

To examine the apoptosis of GBM cells in our microfluidic model, CellEvent Caspase-3/7 Green Detection Reagent (R37111, Thermo Fisher-Scientific) and Hoechst 33342 (5 µg/mL; H3570, Thermo Fisher-Scientific) were used to distinguish the apoptotic cells that with activated caspase-3/7 with bright green nuclei. Briefly, the Caspase-3/7 Detection Reagent was diluted in fresh cell culture media as per the manufacturer's instructions and replenished into the microfluidic chip 3 days after forming the GBM niche. After incubation at 37°C for 1 hr, imaging was conducted immediately by an inverted fluorescent microscope (Zeiss Axio Observer.Z1) with a 10 × objective. The ratio of apoptotic GBM cells were calculated as the number of cells with green nuclei to the total number of GBM cells that were stained with CellTracker Red (5 µM, C34552, Thermo Fisher-Scientific).

## Cytokine quantification

Cytokine concentrations in supernatants were quantified by using human IL-10 (430604, BioLegend), TGF-β1 (88-8350-22, Invitrogen), IFN-γ (430104, BioLegend), and TNF-α (430204, BioLegend) ELISA kits, respectively, according to manufacturer's protocol after collecting supernatants and centrifuging at 2000 × g for 10 min at 4 °C to remove cellular debris. Data was normalized for each type of cytokine using STANDARDIZE function in Excel for *Figure 3* and normalized to control in *Figure 5*.

## Inhibition assays

CSF-1R inhibitor BLZ945 (0.1 µg/ml) and PD-1 blockade nivolumab (1 µg/ml) were used in the monotherapy or dual inhibition treatments. Control groups were treated with fresh cell culture media supplemented with DMSO (0.01%) and human IgG4 isotype control antibody (1 µg/mL, BioLegend). Fresh cell culture media supplemented anti-PD-1 and anti-CSF-1R antibodies was loaded in the microfluidic channels 2 hr after loading allogeneic CD8+ T-cells. Blocking media was freshly prepared and replenished every 24 hr for 3 days.

## Adhesion assay of TAM on HBMVEC capillary

TAMs were obtained by co-culture macrophages with different GBM subtypes using transwell for 3 days. The GBM-educated TAMs were retrieved from transwell and labeled with CellTracker Green (10 µM, C2925, Thermo Fisher-Scientific). HBMVECs were labeled with CellTracker Red (10 µM, C34552, Thermo Fisher-Scientific), and seeded at 100,000 cells/well onto Matrigel pre-coated 24 well-plate (200 µL/well at 37°C for 30 min), and cultured 12 hr to allow for capillary formation. Following the HBMVEC capillary formation, the pre-labeled TAMs were seeded at 100,000 cells/well into the 24 well-plate. Following a 12 hr incubation at 37°C, the unattached TAMs were washed away with warm HBMVEC media for three times. The attached TAMs on HBMVEC capillary were imaged with 20 × objective. The adhered TAMs were then manually counted and plotted as cell number per $10^4$ µm$^2$ HBMVEC area.

## GBM ECM composition analysis

After GBM cells (L08, L20, or L09) and TAMs were cultured in the brain tissue-mimicking hydrogel for 1 and 3 days, different ECM components were fixed and stained with Dylight 488-Laminin (PA522901, Thermo Fisher Scientific), PE-Fibronectin (IC1918P, R and D Systems), and APC-Collagen IV (51-9871-80, Thermo Fisher Scientific) per vendors' instructions. To characterize the HA accumulation, the devices were incubated with Biotinylated Hyaluronic Acid Binding Protein (385911–50 UG, Millipore Sigma) followed by staining with Streptavidin, Alexa Fluor 647 conjugate (S21374, Thermo Fisher Scientific). Each fluorescently stained device was imaged with 6–10 random images using a fluorescent microscope (Zeiss Axio Observer.Z1) with a 40 × objective. The total fluorescent intensity of each image field was quantified using ImageJ (NIH), normalized to its respective DAPI intensity and compared between groups.

## Flow cytometry

To quantify the purity of PBMC-isolated CD8+ cell, cells were washed twice with cold cell sorting buffer containing 2 mM EDTA and 1% BSA in PBS on ice for 10 min. Then, fluorochrome-conjugated antibodies for CD8 (Alexa Fluor 647 anti-human CD8, 344726, BioLegend) were used to label cells for 30 min at 4°C. After washing with 5% BSA buffer solution in PBS twice, labeled cells were measured with a LSRII analyzer (BD Biosciences). All data were analyzed using FlowJo software (Tree Star Inc).

## Confocal microscopy

3D Z-stack images of the engineered GBM microfluidic model were acquired with a Zeiss LSM 710 Laser Scanning Confocal Microscopes with a 20 × objective lens (N.A. = 0.4). To visualize different cell comparts in the model, HBMVEC, GBM cell, and CD8+ T-cell were labeled with DiD (5 μM, V22887, Thermo Fisher-Scientific), CellTracker Red (5 μM, C34552, Thermo Fisher-Scientific) and CellTracker Green (5 μM, C2925, Thermo Fisher-Scientific), respectively prior to loading cells in microfluidics model. All stacked images were reconstructed by using ZEN lite software (Zeiss) for 3D visualization.

## qPCR analysis

RNA was extracted using RNeasy Plus Mini Kit (Qiagen, US), and cDNA was synthesized from the mRNA using SuperScript IV VILO Master Mix for RT-RCR (Invitrogen, US), followed by real-time PCR using a SYBR Green PCR Master Mix (Thermo Fisher Scientific, US) and QuantStudio six sequence detection system (Applied Biosystems, Thermo Fisher Scientific). The reactions were performed with the following cycling conditions: 95°C for 10 min followed by 40 cycles of 95°C for 15 s and 60°C for 1 min. GAPDH was used as a housekeeping gene for normalization. All experiments were repeated three times. The relative change in gene expression was analyzed with the $2^{-\Delta\Delta CT}$ method. The used primers are listed below: CD154 (Forward: CTGATGAAGGGACTTGAC, Reverse: TCTACAGC TTGAACATGC), CD163 (Forward: 5'-CAGGAAACCAGTCCCAAACA-3', Reverse: 5'-AGCGACCTCC TCCATTTACC-3'), PD-1 (Forward: CGTGGCCTATCCACTCCTCA, Reverse: ATCCCTTGTCCCAGC-CACTC), PD-L1 (Forward: 5'- AAATGGAACCTGGCGAAAGC-3', Reverse: 5'- GATGAGCCCC TCAGGCATTT-3'), CSF1 (Forward: GCTGTTGTTGGTCTGTCTC, Reverse: CATGCTCTTCATAATCC TTG), GAPDH (Forward: 5'-GAGTCAACGGATTTGGTCGT-3', Reverse: 5'- TTGATTTTGGAGGGATC TCG-3').

## DNA methylation and data analysis

To assess the epigenetic modifications of TAM in molecularly distinct GBMs, GBM cells and TAMs were co-cultured in the brain tissue-mimicking hydrogel for 3 days, and then recovered using Cell Recovery solution (Corning, NY). TAMs pre-labeled with CellTracker Green (10 μM, C2925, Thermo Fisher-Scientific) and GBM cells were then isolated by using a flow cytometry (MoFlo XDP cell sorter, Beckman Coulter) for DNA methylation. DNA was extracted using automated Maxwell Promega protocol. Whole Genome DNA methylation profiling was performed using Illumina EPIC array, as described previously (*Serrano and Snuderl, 2018*). DNA methylation profiling was performed on all the cases and the raw idats generated from iScan were processed and analyzed using Bioconductor R package Minfi (*Aryee et al., 2014*). All the Illumina array probes were normalized using quantile normalization and corrected for background signal. Samples were then checked for their quality using mean detection p-values and probes with mean detection p-value<0.05 were used for further downstream analysis. Beta values were obtained from the probes that passed the QC as mentioned above. To identify the differentially methylated CpG probes between different groups, dmpFinder function from Minfi package was used. Probes with FDR cutoff (q < 0.05) were considered as most significantly variable probes. Beta value <0.2 means Hypomethylation and Beta value >0.8 means Hypermethylation. For all the differentially methylated CpGs, clustering of samples was analyzed using t-distributed stochastic neighbor embedding (TSNE) method (*Lvd and Hinton, 2008*) that was applied on the 10,000 most variable probes obtained using Minfi package. Heatmaps were generated in a semi-supervised manner using pheatmap package in R, which shows the hierarchical clustering pattern of the top 10,000 significant differentially methylated probes across patients.

## KEGG pathway analysis

For finding the most enriched signaling pathways, we took the most significantly variable probes/genes across groups and passed them through ClusterProfiler (*Yu et al., 2012*) R package for KEGG (Kyoto Encyclopedia of Genes and Genomes) enrichment. The dot plots represent ratio of genes (x-axis) involved in each signaling pathway (y-axis) of KEGG database (*Kanehisa and Goto, 2000*). Size of the dots shows the gene counts and the color denotes the significance level.

## MethylCIBERSORT for immune cell population calculation

To calculate the amount of immune cell population in our cases, we used MethylCIBERSORT (*Chakravarthy et al., 2018*), which is a deconvolution R package used to accurately estimate the cellular composition and tumor purity from DNA methylation data. Beta values obtained from raw idats along with the signature genes were passed through the CIBERSORT (as mentioned in the paper) to deconvolute the immune cell population in our cases.

## Statistical analysis

All data were from at least three independent experiments, and presented as means ± s.e.m (standard error of the mean). The means of groups were compared using one-way analysis of variance (ANOVA) followed by Tukey's post-hoc test in GraphPad Prism or unpaired, two-tailed Student's t-test in Excel (Microsoft), as shown in figure captions. p-Value smaller than 0.05 was considered statistically significant.

## Acknowledgements

This work was supported in part by the US National Institutes of Health (R35GM133646 to WC, R21EB025406 to WC and MS), the National Science Foundation (CBET1701322 to WC), and the Friedberg Charitable Foundation (to MS). DGP was supported by NIH/NINDS (R01NS102665), NIH/NIBIB (R01EB028774), New York State Stem Cell Science (IIRP C32595GG), NYU Grossman School of Medicine, and German Research Foundation (FOR2149).

## Additional information

### Competing interests

Erik P Sulman: Has received consultant fees from Tocagen, Synaptive Medical, Monteris and Robeaute. The other authors declare that no competing interests exist.

### Funding

| Funder | Grant reference number | Author |
|---|---|---|
| National Institute of Biomedical Imaging and Bioengineering | R21EB025406 | Matija Snuderl<br>Weiqiang Chen |
| National Institute of General Medical Sciences | R35GM133646 | Weiqiang Chen |
| National Science Foundation | CBET 1701322 | Weiqiang Chen |
| National Institute of Biomedical Imaging and Bioengineering | R01EB028774 | Dimitris G Placantonakis |
| National Institute of Neurological Disorders and Stroke | R01NS102665 | Dimitris G Placantonakis |
| New York State Stem Cell Science | IIRP C32595GG | Dimitris G Placantonakis |
| NYU Grossman School of Medicine | | Dimitris G Placantonakis |
| German Research Foundation | FOR2149 | Dimitris G Placantonakis |

The funders had no role in study design, data collection and interpretation, or the decision to submit the work for publication.

## Author contributions
Xin Cui, Conceptualization, Data curation, Formal analysis, Validation, Investigation, Visualization, Methodology, Writing - original draft; Chao Ma, Data curation, Formal analysis, Validation, Investigation, Visualization, Writing - review and editing; Varshini Vasudevaraja, Data curation, Formal analysis, Visualization, Methodology; Jonathan Serrano, Jie Tong, Weiyi Qian, Data curation, Formal analysis; Yansong Peng, Michael Delorenzo, Guomiao Shen, Joshua Frenster, Data curation; Renee-Tyler Tan Morales, Formal analysis; Aristotelis Tsirigos, Rajan Jain, Formal analysis, Investigation; Andrew S Chi, Sylvia C Kurz, Erik P Sulman, Resources, Formal analysis, Investigation; Dimitris G Placantonakis, Conceptualization, Resources, Formal analysis, Funding acquisition, Investigation, Methodology, Writing - review and editing; Matija Snuderl, Conceptualization, Data curation, Formal analysis, Supervision, Funding acquisition, Investigation, Visualization, Methodology, Writing - original draft, Project administration; Weiqiang Chen, Conceptualization, Formal analysis, Supervision, Funding acquisition, Investigation, Visualization, Methodology, Writing - original draft, Project administration, Writing - review and editing

## Author ORCIDs
Chao Ma (iD) https://orcid.org/0000-0002-1023-1326
Matija Snuderl (iD) http://orcid.org/0000-0003-0752-0917
Weiqiang Chen (iD) https://orcid.org/0000-0002-9469-8328

## Ethics
Human subjects: Human subjects: Human tumor tissues were harvested from GBM patients undergoing resection surgery of GBM after informed consent. All experimental procedures were reviewed and approved by the NYU Grossman School of Medicine's Institutional Review Board (IRB no.12-01130).

## Decision letter and Author response
Decision letter https://doi.org/10.7554/eLife.52253.sa1
Author response https://doi.org/10.7554/eLife.52253.sa2

# Additional files

## Supplementary files
• Supplementary file 1. Supplementary tables.
• Transparent reporting form

## Data availability
All data generated or analysed during this study are included in the manuscript and supporting files.

The following previously published dataset was used:

| Author(s) | Year | Dataset title | Dataset URL | Database and Identifier |
|---|---|---|---|---|
| Capper D, Jones DTW, Sill M, Hovestadt V | 2018 | DNA methylation-based classification of human central nervous system tumors | https://www.ncbi.nlm.nih.gov/geo/query/acc.cgi?acc=GSE109381 | NCBI Gene Expression Omnibus, GSE109381 |

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
