## [Decision Letter]

**Acceptance summary:**

While the system you describe is very artificial and not validated in an autologous cell setting, the Editors feel that it represents a valuable step in developing more sophisticated GBM-on-a-Chip models.

**Decision letter after peer review:**

Thank you for sending your article entitled "Dissecting the heterogeneity of immunosuppressive microenvironments in glioblastoma-on-a-chip for PD-1 immunotherapy" for peer review at *eLife*. Your article has been evaluated by two peer reviewers, and the evaluation has been overseen by a Reviewing Editor and Jeffrey Settleman as the Senior Editor.

The reviewers and editors agree that it is laudable that you attempt to recreate the tumour microenvironment on a chip. However, there are limitations in such an attempt and the system is felt to be very artificial. The use of PMA-stimulated macrophage cell line and, in particular, of allogeneic CD8+ T cells is of particular concern and it would be important to supplement it with experiments that employ syngeneic PBMC. There are also concerns about the fact that the GBM-on-a-chip model does not take into account blood brain barrier functions and that the data supporting Csf1r as a target need strengthening. Finally, there are methodological details missing that preclude full assessment of the results.

Reviewer #1:

In their manuscript "Dissecting the heterogeneity of immunosuppressive microenvironments in glioblastoma-on-a- chip for PD-1 immunotherapy" Cui et al. describe a novel 'GBM on a chip' to model patient-specific response to PD-1 immunotherapy in 3D microfluidic culture. The authors aim to address a key problem (lack of response to immunotherapy in GBM) using a novel ex vivo microphysiologic culture system. The author use patient-derived GBM cell lines representing the 3 classical sub-types of GBM (based on gene expression profiling), including proneural, classical, and mesenchymal, grown in a PDMS microfluidic device. To mimic the TME, the authors have included several additional cell types, including CD8+ T cells (derived from PBMCs), microvessels (derived from a brain endothelial cell line), and macrophages (derived from U937 monocytic leukemia cell line). The authors conclude their story by demonstrating that CSF1R inhibition represents an attractive strategy to enhance CD8 T cell extravasation and potentially overcome resistance to PD-1 blockade in GBM. The authors present an impressive amount of work, including effort at characterizing epigenetic changes that occur with mono-cultures vs. co-cultures with TAMs (Figure 4).

This study has many novel features worthy of publication and would be of interest to the general *eLife* readership. However, the reviewer cannot support publication of the manuscript in its current form. The manuscript suffers from several major issues (detailed below). With further development, clarification and/or improvement (discussed below), the reviewer believes this study should be reconsidered for publication.

1) Figure 2 – Model tumor microenvironment lacks autologous immune/stromal cells – The 3 additional cell types incorporated into the 'GBM on a chip' are derived from cell lines or unmatched PBMCs. The endothelial cells are brain microvascular endothelial cells. The tumor associated macrophages (TAMs) were generated after PMA stimulation of U937 monocytes. PBMC-derived allogeneic CD8 T cells that were perfused into the 3D brain microvessels are not from the same patient. The authors should emphasize the artificial nature of this model system and highlight the limitations of incorporating multiple cell types from different patients together to 'recapitulate' the TME. The scheme in Supplementary Figure 3A should be in the main part of the manuscript to emphasize the different cell types used in this model. Taken together, this model provides an opportunity to study the migration of various cell types, likely in response to chemokione gradients in a microphysiologic system, but whether or not this recapitulates effector function is unclear.

2) Figure 3 – The data shown in Figure 3 are not very convincing, in contrast to other parts of the study. In addition to questions about the methods used (see #3 below), with the exception of Figure 3G, the changes are modest. The western blots in Figure 3C are of limited quality and difficult to interpret.

3) Materials and methods – The Materials and methods section provide much detail about the method of device fabrication, but very little about the analyses in Figure 3 and 5. The method of quantification is also very confusing. For example, Figure 5H – There are no error bars – how many times was this experiment performed and how many replicates per condition? It is impossible to tell from the data how much cell death exists with each treatment group. An alternative method (e.g. Figure 3—figure supplement 2A) would be recommended.

Reviewer #2:

In the manuscript, the authors used a 'glioblastoma (GBM)-on-a-chip' system based on the microfluidic system and patient-derived cells to study the immunosuppressive tumor microenvironments. The 'GBM-on-a-chip' consisted of the patient-derived GBM cells and tumor-associated macrophages (TAMs) within the hyaluronan-rich Matrigel extracellular matrix (ECM, the middle ring), as well as the brain microvascular and T cells (the outer ring). They monitored the extravasation of CD8+ T cells from the microvessel into the GBM tissue. They reported that mesenchymal GBM, which is associated with the poor response to the immunotherapy, attracted fewer CD154+CD8+ T cells and more TAMs, the M2-like CD68+CD163+ TAMs restricted the CD8+ T-cell recruitment and activation, and PD-1/PD-L1 and immunosuppressive factors were increased in the mesenchymal GBM. Administration of CSF-1R inhibitor reversed the immunosuppressive effects and enhanced the anti-PD-1 efficacy. The 'GBM-on-a-chip' developed a patient-specific therapeutic scanning platform. The study was well-designed and experiments well-performed.

There are some statements needing to be clarified:

1) The brain ECM changes during the tumor development, including HA, collagen, and other components. It is therefore critical to consider the interactions between the cells and ECM in a dynamic manner. As such, the use of HA-rich Matrigel as the ECM should be discussed, and the limitation stated. Perhaps the dynamic changes of the ECM composition as the GBM tumor develops should be characterized.

2) The author did not mention the information of the patients they recruited in the GBM cell culture, such as the sexes, ages, and others, which are also important to understand the GBM development in a patient-specific manner.

3) TAMs and microglia are the major tumor-promoting immune cells in the GBM. Modulation of the TAMs and microglia is considered the promising immunotherapeutic strategy. The TAMs used in this paper were derived from the monocytes – how about the microglia effects?

4) Did the authors evaluate the blood brain barrier functions in this 'GBM-on-a-chip'? There was clearly the vascular component yet its description and discussions on its functions in this particular study were minimum.

5) The title of Figure 2. has a typo 'GBM-a-on-chip'.

---

## [Author Response]

The reviewers and editors agree that it is laudable that you attempt to recreate the tumour microenvironment on a chip. However, there are limitations in such an attempt and the system is felt to be very artificial. The use of PMA-stimulated macrophage cell line and, in particular, of allogeneic CD8+ T cells is of particular concern and it would be important to supplement it with experiments that employ syngeneic PBMC.

We thank the editor and reviewers for pointing out the limitations of our system using macrophage cell lines and allogeneic CD8 T-cells in recapitulating the patient tumor microenvironment. The glioma cell lines have been developed from patients at NYU over several years; however, in our experiments, we did not have access to PBMCs and tumor-associated macrophages from the same patient (which cannot be propagated indefinitely like glioma lines). Thus, as a proof of concept, we used allogeneic CD8+ T-cell isolated from the commercial PBMCs as a replacement of the autologous PBMCs. While imperfect, we believe that the model still recapitulates the interaction between human glioma cells and human immune cells better than current PDX models using immunodeficient mice. In the future, we plan to develop fully personalized chips. In our revised manuscript, we added a discussion to illustrate the limitations of our work on the allogeneic cell resources.

To address this limitation, we planned to collect tumor and stromal cells from the same GBM patient and patient-matched circulating immune cells from peripheral blood and seed all primary cells in a microfluidic chip to validate our findings and combinational therapy in a patient-specific chip. However, due to the recent outbreak of COVID-19 in the New York Area, our access to patient samples are extremely limited. We hope the Editor and reviewer would understand our limited accessibility to patient samples. In the future, we will perform a detailed patient-specific study to further explore the responses of different patient types and mutations.

There are also concerns about the fact that the GBM-on-a-chip model does not take into account blood brain barrier functions and that the data supporting Csf1r as a target need strengthening.

We agree that the intact blood-brain barrier (BBB) can hinder the therapeutically effective drug delivery and limit the drug efficacy in some brain tumors. However, it has been well-documented that the BBB is severely disrupted in the GBM vascular network with leaky abnormal blood vessels leading to contrast enhancement by radiology studies. Therefore, the effect of BBB function on the combinational therapy is limited and we would have to model not a physiological BBB but a barrier disrupted by GBM growth. We added a discussion in our manuscript to discuss the effort of BBB on drug efficacy and the potential limitation of our system without BBB construction. Moreover, in our revised manuscript we present new qPCR analysis data of CSF-1/CSF-1R expression in GBM cell and TAM to support CSF1R as a target in combinational therapy.

Finally, there are methodological details missing that preclude full assessment of the results.

We have updated the Materials and methods section with more details of our experiments.

Reviewer #1:[…] This study has many novel features worthy of publication and would be of interest to the general eLife readership. However, the reviewer cannot support publication of the manuscript in its current form. The manuscript suffers from several major issues (detailed below). With further development, clarification and/or improvement (discussed below), the reviewer believes this study should be reconsidered for publication.1) Figure 2 – Model tumor microenvironment lacks autologous immune/stromal cells – The 3 additional cell types incorporated into the 'GBM on a chip' are derived from cell lines or unmatched PBMCs. The endothelial cells are brain microvascular endothelial cells. The tumor associated macrophages (TAMs) were generated after PMA stimulation of U937 monocytes. PBMC-derived allogeneic CD8 T cells that were perfused into the 3D brain microvessels are not from the same patient. The authors should emphasize the artificial nature of this model system and highlight the limitations of incorporating multiple cell types from different patients together to 'recapitulate' the TME.

We thank the reviewer for this essential suggestion. In our experiments, we used U937 monocyte cell line and allogeneic CD8^+^ T-cell and brain microvascular endothelial cell due to the lack of access to the autologous immune/stromal cells from the same patients. Following the reviewer’s suggestion, we have added a discussion in the third paragraph of the Discussion section to illustrate the artificial nature and note the limitations of our current model system using the allogeneic cell resources.

The scheme in Supplementary Figure 3A should be in the main part of the manuscript to emphasize the different cell types used in this model.

We thank the reviewer for this suggestion. In our revised manuscript, we moved the scheme in Supplementary Figure 3A to the main Figure 2B to illustrate the protocol of the microphysiological system.

Taken together, this model provides an opportunity to study the migration of various cell types, likely in response to chemokione gradients in a microphysiologic system, but whether or not this recapitulates effector function is unclear.

The T-cell and TAM infiltration and migration behaviors in the GBM microenvironment are important factors that affect immunotherapy efficacy. Hence, in the revised manuscript we added new results of effector function assays (such as perforin and granzyme B) to validate the T-cell activation (Figure 3—figure supplement 1) and the cytolytic activity of CD8^+^ T-cell with GBM cell apoptosis assay (Figure 2J). We also added a discussion in the subsection “Distinct extravasation and cytotoxic activities of allogeneic CD8+ T-cells in GBM subtypes” to highlight how our model can recapitulate effector function of CD8^+^ T-cell in the tumor microenvironments.

2) Figure 3 – The data shown in Figure 3 are not very convincing, in contrast to other parts of the study. In addition to questions about the methods used (see #3 below), with the exception of Figure 3G, the changes are modest. The western blots in Figure 3C are of limited quality and difficult to interpret.

We thank the reviewer to raise the concern about the Figure 3. To improve this, we conducted additional experiments to improve the data quality of all results in Figure 3. We have done more repeats of experiments and updated Figure 3B to show the significant difference in different markers among the molecularly-different GBMs. We replaced the western blots result with new qPCR analysis data showing different PD-1 and CD154 expressions in CD8^+^ T-cell, and PD-L1, CSF-1/CSF-1R expressions in GBM cell and TAM (Figure 3C, Figure 5—figure supplement 1). Furthermore, we revised the results description to illustrate the quantification methods in more details.

3) Materials and methods – The Materials and methods section provide much detail about the method of device fabrication, but very little about the analyses in Figures 3 and 5.

We have revised the manuscript to illustrate the quantification methods for the analyses in Figures 3 and 5 in more details.

The method of quantification is also very confusing. For example, Figure 5H – There are no error bars – how many times was this experiment performed and how many replicates per condition? It is impossible to tell from the data how much cell death exists with each treatment group. An alternative method (e.g. Figure 3—figure supplement 2A) would be recommended.

We think the confusing is mainly induced by the missing of detailed experiment parameters and quantification methods used in Figure 5H. The data has error bars as they are from at least three independent experiments for each condition. To compare the difference between control and drug (PD-1, CSF1R and dual inhibitors)–treated groups for the same GBM patients’ derived cell line, we calculated the relative percentage of mean apoptosis ratio in different groups. To make it clear, we have updated this figure by adding error bars for each experimental group, provide the absolute apoptosis data in new supplementary figures (Figure 5—figure supplement 3), and illustrate the details in the Materials and methods.

Reviewer #2:[…] There are some statements needing to be clarified:1) The brain ECM changes during the tumor development, including HA, collagen, and other components. It is therefore critical to consider the interactions between the cells and ECM in a dynamic manner. As such, the use of HA-rich Matrigel as the ECM should be discussed, and the limitation stated. Perhaps the dynamic changes of the ECM composition as the GBM tumor develops should be characterized.

We agree with the reviewer that the dynamic interactions between the GBM and ECM are essential for tumor growth. Therefore, we performed additional experiments to investigate the dynamic changes of the ECM composition interacting with molecularly-different GBM cells. Composition of ECM in MES (GBML91), CL (GBML08) and PN (GBML20) GBM niches were examined with ECM antibodies (such as collagen VI, laminin, hyaluronan and fibronectin) to determine the matrix components in Figure 4—figure supplement 3, but showed no significant difference in the three-day culture period. It might be because a short-term culture and predefined ECM composition might not be able to reflect the actual in vivo dynamic interactions of GBM and ECM. We highlighted the importance of the ECM used in the current study in the third paragraph of the Discussion.

2) The author did not mention the information of the patients they recruited in the GBM cell culture, such as the sexes, ages, and others, which are also important to understand the GBM development in a patient-specific manner.

We added more information about these patients in the Supplementary file 1.

3) TAMs and microglia are the major tumor-promoting immune cells in the GBM. Modulation of the TAMs and microglia is considered the promising immunotherapeutic strategy. The TAMs used in this paper were derived from the monocytes – how about the microglia effects?

To illustrate the effects of microglia, we conducted additional experiments including human microglia in our system to investigate whether microglia differently affects the immunosuppressive microenvironment and the therapy efficiency (Figure 5—figure supplement 4). Our results indicated that microglia in the GBM microenvironment might have a similar immunosuppressive effect with PBMC-derived macrophages on CD8+ T-cell PD-1 expression and functionality. Yet, no significant change in GBM cell apoptosis response to the PD-1 and CSF-1R dual blockade treatment was observed in our study, which might be contributed by the complex reprogramming of microglia phenotypes in the brain tumor microenvironment. We added the results in the subsection “Optimizing anti-PD-1 therapy by co-targeting TAM CSF-1 signaling” and discussion in the third paragraph of the Discussion section.

4) Did the authors evaluate the blood brain barrier functions in this 'GBM-on-a-chip'? There was clearly the vascular component yet its description and discussions on its functions in this particular study were minimum.

Intact blood-brain barrier (BBB) can hinder the therapeutically effective drug delivery and limit the drug efficacy in some brain tumors. However, it is well-established that BBB is uniformly disrupted in GBM which leads to leaky blood vessels (Sarkaria et al., 2018). Studying the effect on disrupted BBB in GBM is beyond the scope of this study. Hence, in this study we would like to focus on the immune cell activity, rather than the BBB aberrations in GBM. Nevertheless, our simplified GBM microenvironment model without BBB construction, although might not perfectly mimic the in vivo structure, can still serve as a suitable and useful model to dissect the GBM tumor-immune-vascular interactions ex vivo. We revised our manuscript and discussed the weakness of our current model on mimicking the BBB functions in GBM in the third paragraph of the Discussion section.

5) The title of Figure 2. has a typo 'GBM-a-on-chip'.

We have corrected the typo and double-check the entire manuscript again for other typos in the manuscript.